# Evaluation of the reliability and risks of ChatGPT-4o in answering pediatric cough questions: A comparative analysis between pediatricians and pediatric pulmonologists

Hanife Tuğçe Çağlar[1]*, Emine Özdemir Kaçer[2], Sevgi Pekcan[1], Fatma Nur Ayman[1]

1 Department of Pediatric Pulmonology, Necmettin Erbakan University School of Medicine, Konya, Turkey,
2 Department of Pediatrics, Aksaray University School of Medicine, Aksaray, Turkey

* h.tugce.s@gmail.com

## Abstract

### Introduction

Artificial intelligence tools such as ChatGPT are increasingly used by patients and healthcare professionals, yet their reliability in pediatric respiratory conditions remains unclear. This study aims to assess the trustworthiness, comprehensiveness, value, and potential dangers of ChatGPT-4o-generated responses to frequently asked questions about the management and care of cough in children.

### Materials and methods

A total of 10 cough-related questions were selected for ChatGPT-4o. The questions and responses generated by ChatGPT-4o are presented to 32 pediatric pulmonologists and 32 pediatricians. An online questionnaire was developed for this study. Participants rated the answers generated by ChatGPT-4o on a scale of 1–10 in terms of trustworthiness, comprehensiveness, value, and danger. Higher scores indicate higher levels of trustworthiness, comprehensiveness, value and danger. In addition, a yes/no question asked participants if there was anything wrong with the answer generated by ChatGPT-4o.

### Results

The ChatGPT-4o-generated answers were generally rated by participants as trustworthy (median:6.45, IQR:1.97), valuable (median:6.15, IQR:2.30), comprehensive (median:6.15, IQR:1.83), and not dangerous (median:4.35, IQR:2.65). There was a statistically significant difference in all overall ratings between pulmonologists and pediatricians. Pediatricians rated ChatGPT-4o-generated answers as more trustworthy, valuable, comprehensive, and less dangerous compared to pediatric pulmonologists. For each of the ten questions, at least one participant indicated that there

**Citation:** Çağlar HT, Özdemir Kaçer E, Pekcan S, Ayman FN (2025) Evaluation of the reliability and risks of ChatGPT-4o in answering pediatric cough questions: A comparative analysis between pediatricians and pediatric pulmonologists. PLoS One 20(12): e0340007. https://doi.org/10.1371/journal.pone.0340007

**Data availability statement:** All relevant data are within the manuscript and its Supporting information files.

**Funding:** The author(s) received no specific funding for this work.

**Competing interests:** The authors have declared that no competing interests exist.

was something wrong with the ChatGPT-4o-generated response. However, for no question did the proportion of "yes" responses exceed 50%, indicating that concerns were not universally shared among participants.

## Conclusion

Our study highlights both the potential benefits and limitations of ChatGPT-4o in providing medical information about pediatric cough. While AI-generated responses were generally rated as trustworthy and valuable, differences in assessment between pediatricians and pediatric pulmonologists emphasize the need for careful interpretation of AI-derived medical content.

## 1. Introduction

Coughing is a protective reflex that plays an important role in clearing secretions and foreign material from the respiratory tract [1]. Cough is one of the most common symptoms of respiratory disease and a leading cause of hospitalization in children [2]. Cough in children is often associated with viral infections, most of which usually resolve spontaneously. Cough following influenza infection; may last up to 10 days in 35–40% of school-aged children and up to 25 days in 10% of preschool children following respiratory tract infection. Bronchial hyperactivity, asthma, and gastroesophageal reflux disease are other common causes of cough in children [3]. In addition, environmental factors such as indoor and outdoor air pollution, humidity, irritant gases, and cigarette smoking cause cough in children [4].

If the cough persists for a long time, it becomes very distressing. It affects sleep, daily activities, and quality of life for the child and the parents or caregivers. Parents want the cough to go away immediately and search for solutions [5]. Online resources have become the first source of health information for many patients, allowing them to learn about their health condition [6]. The term "Dr. Google" refers to patients' use of the Internet to search for health information [7]. Google® is the most popular search engine today and the most visited website in the world [8].

ChatGPT, a cutting-edge language model developed by Open Artificial Intelligence (OpenAI), based on the Generative Pre-trained Transformer (GPT) series [9], has demonstrated outstanding performance in natural language processing tasks that require the generation of coherent, contextually relevant, and human-like responses. ChatGPT, currently the fastest-growing consumer application, has shown increasing potential in medical education, research, and healthcare delivery [10–12]. This potential has been demonstrated in several medical fields such as radiology and dermatology [13,14]. It has the potential to assist individuals and communities in making informed decisions about their health, with its ability to generate human-like text based on large amounts of data [15–17]. However, a common criticism of ChatGPT is that the generated text is not always accurate. Challenges in this area have arisen in the areas of data quality and diversity, explainability and trust, and regulatory and ethical considerations of AI [18–20]. This study aims to assess the trustworthiness,

comprehensiveness, value, and potential dangers of ChatGPT-4o-generated responses to frequently asked questions about the management and care of cough in children.

## 2. Materials and methods

### 2.1. Study design

An initial search was conducted on Google® to identify the "most frequently asked questions about pediatric coughing." To reduce algorithmic bias, web browsing history and cookies were cleared prior to the search. Questions were excluded if they were semantically redundant, vague, subjective, or non-medical in nature. The final selection of 10 questions was reviewed by two pediatric pulmonologists (authors) to ensure clinical relevance. The final set of 10 questions was chosen because they consistently appeared across multiple independent searches and represented the most commonly encountered parent-driven concerns in outpatient pediatric respiratory practice. Ten unique, medically relevant questions related to coughing were submitted to ChatGPT-4o. The corresponding AI-generated responses are provided in the Supplementary Material.

In addition to the structured Likert questions, we considered open-ended questions during study planning. However, to minimize participant burden and ensure higher response rates among geographically dispersed physicians, we opted for a fully standardized questionnaire. This design enabled consistent quantitative comparisons between specialties. We acknowledge the limitation of not including open-ended responses, as they could have provided richer context.

We selected ChatGPT-4o because, at the time the study began, it was the most widely used large language model accessible to the general public. It also had multilingual capabilities relevant to the study population and demonstrated superior performance in several medical information tasks in recent evaluations. Using a widely adopted model also increases the real-world relevance of our findings.

The study was conducted in accordance with the latest version (2013) of the Declaration of Helsinki and approved by the Necmettin Erbakan University Ethics Committee with decision number (2025/5864).

A total of 100 physicians (50 pediatricians and 50 pediatric pulmonologists) were invited to participate. Sixty-four (32 pediatric pulmonologists and 32 pediatricians) of them completed the survey, yielding a response rate of 75.3%. Recruitment for the study began on June 28, 2025, and ended on July 5, 2025. The participating physicians had an age range of 30–65 years and reported between 5 and 35 years of clinical experience. All participants were practicing in various regions across Türkiye, representing urban healthcare settings. An online questionnaire was developed for this study. Participants rated the AI-generated responses on a 10-point Likert scale across four dimensions: trustworthiness, comprehensiveness, value, and potential danger. Higher scores indicate higher levels of trustworthiness, comprehensiveness, value and potential danger. Although the Likert scale provides quantifiable measures of trustworthiness, value, comprehensiveness, and danger, it is inherently subjective. To mitigate this, we included a binary (yes/no) question asking whether there was anything clinically wrong in each ChatGPT-4o response. Because the Likert items measured straightforward constructs (trustworthiness, comprehensiveness, value, and danger) and were developed by two pediatric pulmonologists, content and face validity were ensured through expert review. However, formal psychometric validation (e.g., Cronbach's alpha) was not performed because the four items measured distinct constructs intentionally. Written informed consent was obtained from the participants and no compensation was paid. Participants were allowed to stop completing the questionnaire at any time. The questionnaire contained no identifying information, and the data were confidential.

In this study, the responses generated by ChatGPT-4o were evaluated against the clinical judgment and expertise of pediatricians and pediatric pulmonologists, who served as the reference standard. Pediatric pulmonologists, in particular, were considered a benchmark due to their advanced training and specialization in respiratory diseases in children. While acknowledging that AI tools may occasionally outperform human judgment in certain tasks, this study positions physicians' assessments as the current clinical gold standard for evaluating the medical appropriateness and safety of the information provided.

## 2.2. Statistical analysis

As the study utilized Likert-type responses (1–10 scale), data were summarized using medians and interquartile ranges (IQR). Categorical data are presented as frequencies and percentages. Differences in categorical variables were assessed using the chi-squared test, while comparisons of numerical (Likert-type) responses between groups were performed using the Mann–Whitney U test. Data were analyzed using SPSS version 22.0 (SPSS Inc, Chicago, IL, USA). A p-value of <0.05 was considered statistically significant.

Additionally, a post hoc power analysis was performed using G*Power 3.1 based on the observed between-group differences in the overall Likert scores. With 32 pediatricians and 32 pediatric pulmonologists, the study had adequate statistical power (>80%) to detect the observed effect sizes for the trustworthiness, value, comprehensiveness, and danger ratings (Cohen's d was approximately 0.82–0.98, and the achieved power was 0.82–0.88 at $\alpha = 0.05$).

## 3. Results

The ChatGPT-4o-generated answers were generally rated by participants as trustworthy (median: 6.45, IQR:1.97), valuable (median: 6.15, IQR:2.3), comprehensiveness (median: 6.15, IQR:1.83), and not dangerous (median: 4.35, IQR:2.65). There was a statistically significant difference in all overall ratings between pulmonologists and pediatricians. All but two of the ten responses received median scores of 5 or higher from participants. The overall ratings of ten answers are summarized in Table 1. The distribution of the Likert scores between pediatricians and pediatric pulmonologists is illustrated in Fig 1.

Pediatricians rated ChatGPT-4o-generated answers as more trustworthy, valuable, comprehensive, and less dangerous compared to pediatric pulmonologists. Table 2 summarizes the comparison between pediatricians and pediatric pulmonologists for each ChatGPT-4o-generated answer.

Among the ten questions evaluated, "What are the different types of cough in children?" and "When does a child need antibiotics for a cough?" received the highest overall ratings for trustworthiness and value across both groups. In contrast, "How do you stop a child from coughing at night?" and "What are three common causes of cough?" were rated the lowest, particularly by pediatric pulmonologists. Notably, significant discrepancies between pediatricians and pediatric pulmonologists were observed in several questions. For instance, the largest divergence in danger ratings was seen in the question about stopping a child from coughing at night, where pulmonologists assigned a significantly higher danger score (median:8.00, IQR:3) compared to pediatricians (median:4.00, IQR:5). These results indicate not only varying perceptions of AI reliability across subspecialties but also the importance of clinical nuance in interpreting AI-provided advice.

For all ten questions, at least one participant answered "yes" to the question, "Is there anything wrong with the answer generated by ChatGPT-4o?". This indicates that every AI-generated response raised concern for at least one physician. However, none of the questions received "yes" responses from more than half of the participants, suggesting that perceived issues were neither consistent nor consensus-based. Table 3 summarizes the comparison between pediatricians and pediatric pulmonologists for this yes/no question.

**Table 1. The overall ratings of the ChatGPT-4o-generated answers.**

|  | All participants (n = 64) | Pediatricians (n = 32) | Pediatric pulmonologists (n = 32) | p value | Effect size r (95% CI) |
|---|---|---|---|---|---|
| Trustworthiness | 6.45 (1.97) | 7.15 (2.30) | 6.00 (1.95) | **0.001** | **0.43 (0.20–0.65)** |
| Value | 6.15 (2.30) | 6.90 (2.45) | 5.55 (2.18) | **0.002** | **0.38 (0.15–0.61)** |
| Comprehensiveness | 6.15 (1.83) | 6.50 (2.00) | 5.40 (1.95) | **<0.001** | **0.44 (0.22–0.67)** |
| Danger | 4.35 (2.65) | 3.30 (2.90) | 4.85 (2.35) | **0.002** | **0.39 (0.16–0.62)** |

All data are expressed as median (IQR). Effect sizes (r) were calculated for the Mann–Whitney U tests as $r = Z/\sqrt{N}$, and 95% confidence intervals were derived from the standard error of r.

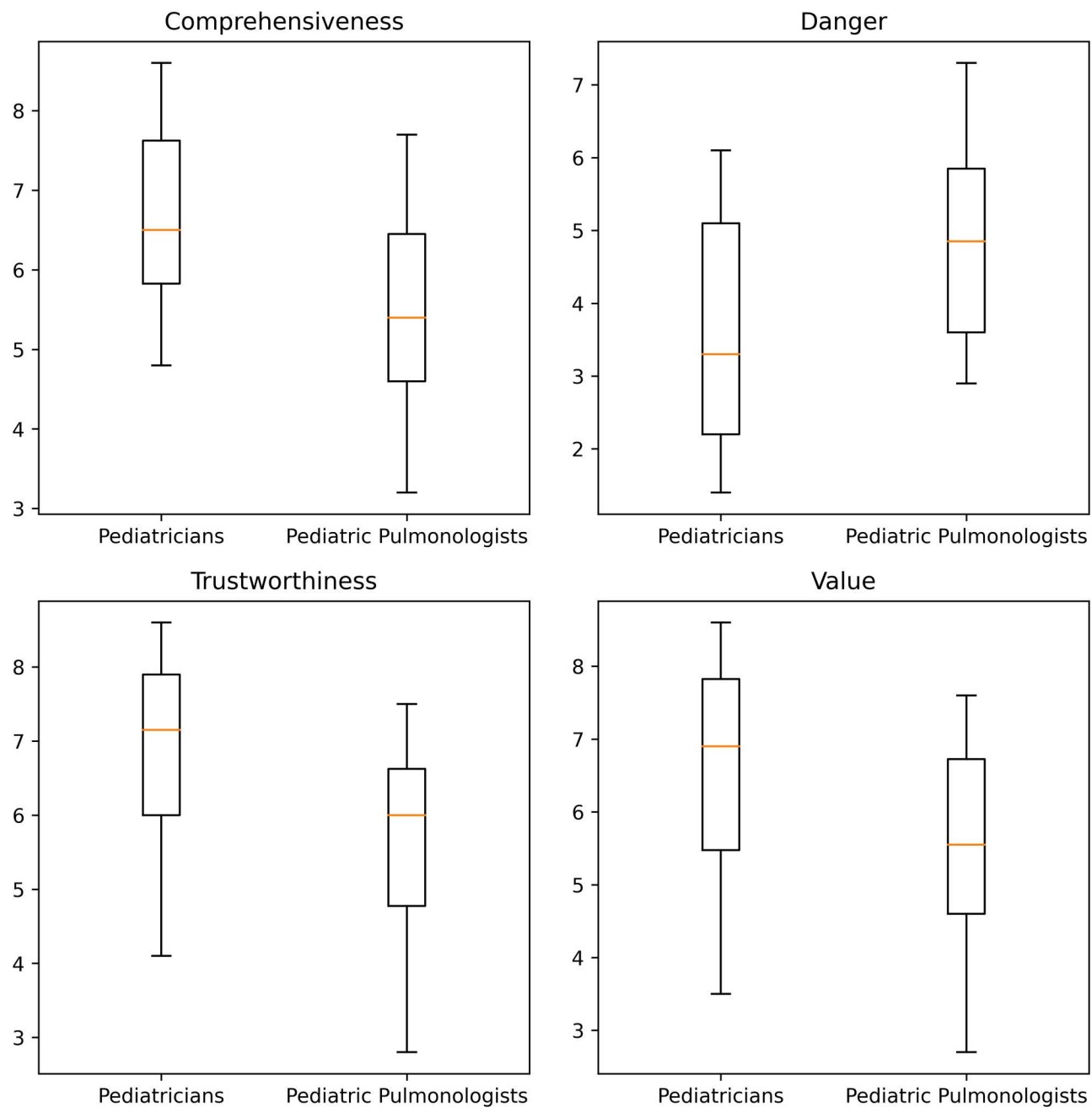

**Fig 1. Boxplots comparing pediatricians and pediatric pulmonologists for trustworthiness, value, comprehensiveness, and danger ratings.**

## 4. Discussion

This study assessed the trustworthiness, comprehensiveness, value, and potential dangers of ChatGPT-4o-generated responses to common questions regarding the management and care of cough in children. Our findings indicate that, overall, ChatGPT-4o-generated responses were perceived as trustworthy, valuable, and comprehensive, with median scores exceeding 5 in most cases. However, a notable variation was observed between general pediatricians and

**Table 2. The comparison between pediatricians and pediatric pulmonologists for each ChatGPT-4o-generated answer.**

| | | Pediatricians (n = 32) | Pediatric pulmonologists (n = 32) | p value | Effect size r (95% CI) |
|---|---|---|---|---|---|
| 1. What are the different types of cough in children? | 1a. Trustworthiness | 8.50 (1) | 8.00 (2) | 0.100 | 0.206 (−0.45 - 0.04) |
| | 1b. Value | 9.00 (1) | 8.00 (2) | 0.167 | 0.173 (−0.42 - 0.07) |
| | 1c. Comprehensiveness | 8.00 (2) | 8.00 (2) | 0.161 | 0.175 (−0.42 - 0.07) |
| | 1d. Danger | 3.00 (1) | 3.00 (2) | 0.894 | 0.017 (−0.26 - 0.23) |
| 2. When does a child need antibiotics for a cough? | 2a. Trustworthiness | 8.00 (2) | 8.00 (3) | 0.193 | 0.163 (−0.41 - 0.08) |
| | 2b. Value | 8.00 (3) | 8.00 (3) | 0.461 | 0.092 (−0.34 - 0.15) |
| | 2c. Comprehensiveness | 8.00 (2) | 7.00 (3) | 0.141 | 0.184 (−0.43 - 0.06) |
| | 2d. Danger | 3.00 (2) | 3.50 (2) | 0.389 | 0.108 (−0.35–0.14) |
| 3. When should you worry about a child's cough? | 3a. Trustworthiness | 7.00 (2) | 5.00 (2) | **<0.001** | **0.547 (0.34–0.75)** |
| | 3b. Value | 7.00 (3) | 5.00 (1) | **0.002** | **0.390 (0.16–0.62)** |
| | 3c. Comprehensiveness | 6.00 (3) | 5.00 (2) | **0.006** | **0.342 (0.11–0.57)** |
| | 3d. Danger | 4.00 (3) | 5.00 (2) | **0.002** | **0.392 (0.16–0.62)** |
| 4. What is patient education for cough in children? | 4a. Trustworthiness | 8.00 (4) | 7.50 (3) | 0.215 | 0.155 (−0.40–0.09) |
| | 4b. Value | 7.50 (3) | 7.00 (3) | 0.307 | 0.128 (−0.37–0.12) |
| | 4c. Comprehensiveness | 8.00 (3) | 8.00 (2) | 0.483 | 0.088 (−0.33–0.16) |
| | 4d. Danger | 2.00 (3) | 3.00 (4) | 0.475 | 0.089 (−0.33–0.15) |
| 5. What are three common causes of cough? | 5a. Trustworthiness | 4.00 (2) | 3.50 (3) | **0.006** | **0.345 (0.12–0.57)** |
| | 5b. Value | 4.00 (3) | 3.00 (3) | **0.007** | **0.338 (0.11–0.57)** |
| | 5c. Comprehensiveness | 4.00 (1) | 2.00 (3) | **<0.001** | **0.451 (0.23–0.67)** |
| | 5d. Danger | 5.50 (4) | 7.00 (2) | **0.017** | **0.298 (0.07–0.52)** |
| 6. What are cough warning signs? | 6a. Trustworthiness | 8.00 (3) | 6.00 (3) | **<0.001** | **0.497 (0.29–0.70)** |
| | 6b. Value | 8.00 (4) | 6.00 (3) | **<0.001** | **0.459 (0.25–0.67)** |
| | 6c. Comprehensiveness | 8.00 (3) | 6.00 (3) | **<0.001** | **0.540 (0.34–0.75)** |
| | 6d. Danger | 2.50 (4) | 4.00 (5) | **0.025** | **0.281 (0.06–0.50)** |
| 7. How do you stop a child from coughing at night? | 7a. Trustworthiness | 5.00 (3) | 2.50 (3) | **<0.001** | **0.613 (0.42–0.81)** |
| | 7b. Value | 5.00 (4) | 2.00 (3) | **<0.001** | **0.579 (0.38–0.78)** |
| | 7c. Comprehensiveness | 5.00 (4) | 2.00 (2) | **<0.001** | **0.530 (0.32–0.74)** |
| | 7d. Danger | 4.00 (5) | 8.00 (3) | **<0.001** | **0.602 (0.41–0.80)** |
| 8. What is the best treatment for cough in child? | 8a. Trustworthiness | 8.00 (2) | 6.00 (3) | **<0.001** | **0.483 (0.27–0.70)** |
| | 8b. Value | 8.00 (2) | 6.00 (3) | **<0.001** | **0.452 (0.23–0.67)** |
| | 8c. Comprehensiveness | 8.00 (2) | 6.00 (3) | **<0.001** | **0.548 (0.34–0.75)** |
| | 8d. Danger | 2.50 (4) | 4.50 (4) | **0.001** | **0.427 (0.20–0.65)** |
| 9. What is a croup cough? | 9a. Trustworthiness | 7.00 (1) | 6.00 (3) | **0.002** | **0.393 (0.17–0.62)** |
| | 9b. Value | 7.00 (1) | 6.00 (3) | **0.005** | **0.352 (0.12–0.58)** |
| | 9c. Comprehensiveness | 7.00 (1) | 6.00 (3) | **<0.001** | **0.479 (0.26–0.70)** |
| | 9d. Danger | 2.50 (4) | 5.00 (3) | **0.001** | **0.431 (0.21–0.65)** |
| 10. What to do with a child coughing while sleeping? | 10a. Trustworthiness | 7.00 (4) | 7.00 (1) | 0.116 | 0.197 (−0.44–0.05) |
| | 10b. Value | 7.00 (4) | 6.00 (4) | 0.068 | 0.228 (−0.47–0.01) |
| | 10c. Comprehensiveness | 7.00 (2) | 7.00 (3) | 0.513 | 0.082 (−0.33–0.16) |
| | 10d. Danger | 3.00 (4) | 4.00 (5) | 0.203 | 0.159 (−0.40–0.09) |

All data are expressed as median (IQR). Effect sizes (r) were calculated for the Mann–Whitney U tests as $r = Z/\sqrt{N}$, and 95% confidence intervals were derived from the standard error of r.

**Table 3. The comparison between pediatricians and pediatric pulmonologists for each ChatGPT-4o-generated answer. "Is there anything wrong with the answer generated by ChatGPT-4o?".**

| | | Pediatricians (n = 32) | Pediatric pulmonologists (n = 32) | *p* value |
|---|---|---|---|---|
| 1. What are the different types of cough in children? | Yes | 4 (12.5) | 11 (34.4) | **0.039** |
| 2. When does a child need antibiotics for a cough? | Yes | 4 (12.5) | 9 (28.1) | 0.120 |
| 3. When should you worry about a child's cough? | Yes | 1 (3.1) | 10 (31.3) | **0.003** |
| 4. What is patient education for cough in children? | Yes | 6 (18.8) | 12 (37.5) | 0.095 |
| 5. What are three common causes of cough? | Yes | 3 (9.4) | 8 (25) | 0.098 |
| 6. What are cough warning signs? | Yes | 2 (6.3) | 9 (28.1) | **0.020** |
| 7. How do you stop a child from coughing at night? | Yes | 3 (9.4) | 8 (25) | 0.098 |
| 8. What is the best treatment for cough in child? | Yes | 6 (18.8) | 13 (40.6) | 0.055 |
| 9. What is a croup cough? | Yes | 2 (6.3) | 9 (28.1) | **0.020** |
| 10. What to do with a child coughing while sleeping? | Yes | 4 (12.5) | 10 (31.3) | 0.129 |

All data are expressed as n (%).

pediatric pulmonologists in their evaluations, highlighting differences in expectations and clinical perspectives regarding AI-generated medical information.

One of the key findings of this study is that pediatricians rated ChatGPT-4o-generated responses more favorably in terms of trustworthiness, value, and comprehensiveness compared to pediatric pulmonologists. This may reflect a difference in the level of clinical expertise and familiarity with specialized aspects of respiratory diseases. Pediatric pulmonologists, having a deeper knowledge of complex respiratory conditions, may have been more critical in their assessments, particularly regarding comprehensiveness and the potential for misinformation. Pediatric pulmonologists may have rated the responses more critically because they routinely manage complex, atypical, and high-risk respiratory cases. Their training increases their sensitivity to nuances, red flags, and rare differential diagnoses. This makes them more likely to detect omissions or oversimplifications in AI-generated content. Some disagreement among physicians may reflect ongoing clinical controversies or differences in interpretation, rather than clear-cut errors in the AI-generated text. This highlights the complexity of defining a single "correct" answer in certain pediatric scenarios. These findings align with previous research indicating that AI-generated medical responses can be useful but may require domain-specific validation before clinical implementation [21,22].

Despite the generally positive ratings, concerns regarding the accuracy and comprehensiveness of AI-generated responses remain. At least one participant flagged an issue with every ChatGPT-4o-generated response, though the number of "yes" responses did not exceed 50% for any question. This underscores the importance of human oversight in AI-generated health information. AI models like ChatGPT-4o have demonstrated impressive linguistic fluency and knowledge synthesis capabilities, but they may still produce responses that lack nuance or fail to consider individual patient contexts [23]. Furthermore, AI-generated content may not always reflect the latest medical guidelines, which is a significant limitation in the rapidly evolving field of pediatric medicine [24]. While it is true that AI-generated content may not always reflect the latest clinical guidelines, it is equally important to recognize that not all physicians are consistently up to date with evolving standards of care. Continuing medical education varies between practitioners, and even specialists may occasionally rely on outdated or incomplete information. Therefore, discrepancies between AI responses and expert opinions could stem from limitations in either source. This observation further supports the need for AI tools to serve as complementary aids rather than replacements for professional medical judgment.

Another critical aspect explored in this study is the potential risks associated with AI-generated medical information. While ChatGPT-4o responses were generally not considered dangerous, some variability in danger ratings was observed.

Pediatric pulmonologists tended to assign higher danger scores compared to general pediatricians, possibly due to their awareness of subtle clinical nuances and the potential consequences of misinformation in complex respiratory cases. This highlights the necessity of ensuring that AI-generated health information is reviewed by medical professionals and supplemented with expert validation before being used in clinical practice.

The growing reliance on AI-based tools such as ChatGPT-4o for medical information raises important ethical and practical concerns. The accessibility of AI-generated medical advice can empower patients and caregivers by improving health literacy and facilitating informed decision-making. However, it also poses risks, such as the spread of misinformation, over-reliance on AI in place of professional medical consultation, and challenges in ensuring accountability for AI-generated recommendations. Regulatory frameworks and quality assurance measures should be developed to enhance the reliability of AI in healthcare and mitigate potential risks associated with its widespread use.

This study has several limitations. The evaluation was conducted by medical professionals rather than the general public, which may not fully capture how non-medical users perceive and interpret AI-generated responses. The Likert scale was not formally validated, which could impact its internal consistency. However, the constructs were intentionally kept independent, and the items underwent expert review to maximize content validity. The study assessed a limited number of questions related to pediatric cough management, and findings may not be generalizable to other areas of medicine. AI models continue to evolve, and future versions of ChatGPT may demonstrate improved accuracy and reliability in medical applications. Although the yes/no question identified that at least one participant found issues in each answer, we did not ask participants to explain the reasons behind their "yes" responses. Without qualitative explanations, it is difficult to interpret these concerns. Future studies should incorporate open-ended follow-up questions to better understand why physicians judge certain AI-generated statements as problematic. The generalizability of the findings may be limited by the homogeneity of the physician population and the cultural-linguistic context.

## 5. Conclusion

Our study highlights both the potential benefits and limitations of ChatGPT-4o in providing medical information about pediatric cough. While AI-generated responses were generally rated as trustworthy and valuable, differences in assessment between pediatricians and pediatric pulmonologists emphasize the need for careful interpretation of AI-derived medical content. Future research should aim to refine AI algorithms for greater medical accuracy, while also exploring how both healthcare professionals and laypeople interpret and respond to AI-generated content across diverse settings. Incorporating qualitative feedback and broader participant profiles will help ensure the safe and effective integration of AI into clinical practice.

## Supporting information

**S1 File. The corresponding AI-generated responses.**
(DOCX)

**S2 File. Dataset.**
(RAR)

## Author contributions

**Conceptualization:** Hanife Tuğçe Çağlar, Emine Özdemir Kaçer, Sevgi Pekcan, Fatma Nur Ayman.

**Data curation:** Hanife Tuğçe Çağlar, Emine Özdemir Kaçer, Sevgi Pekcan, Fatma Nur Ayman.

**Formal analysis:** Hanife Tuğçe Çağlar, Emine Özdemir Kaçer, Sevgi Pekcan, Fatma Nur Ayman.

**Funding acquisition:** Hanife Tuğçe Çağlar, Emine Özdemir Kaçer, Sevgi Pekcan, Fatma Nur Ayman.

**Investigation:** Hanife Tuğçe Çağlar, Emine Özdemir Kaçer, Sevgi Pekcan, Fatma Nur Ayman.

**Methodology:** Hanife Tuğçe Çağlar, Emine Özdemir Kaçer, Sevgi Pekcan, Fatma Nur Ayman.

**Project administration:** Hanife Tuğçe Çağlar, Emine Özdemir Kaçer, Sevgi Pekcan, Fatma Nur Ayman.

**Resources:** Hanife Tuğçe Çağlar, Emine Özdemir Kaçer, Sevgi Pekcan, Fatma Nur Ayman.

**Software:** Hanife Tuğçe Çağlar, Emine Özdemir Kaçer, Sevgi Pekcan.

**Supervision:** Hanife Tuğçe Çağlar, Emine Özdemir Kaçer, Sevgi Pekcan.

**Validation:** Hanife Tuğçe Çağlar, Emine Özdemir Kaçer, Sevgi Pekcan.

**Visualization:** Hanife Tuğçe Çağlar, Emine Özdemir Kaçer, Sevgi Pekcan, Fatma Nur Ayman.

**Writing – review & editing:** Emine Özdemir Kaçer, Fatma Nur Ayman.

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
