## [Decision Letter · Decision Letter 0]

10 Nov 2025

Dear Dr. Çağlar,

Thank you for submitting your manuscript to PLOS ONE. After careful consideration, we feel that it has merit but does not fully meet PLOS ONE’s publication criteria as it currently stands. Therefore, we invite you to submit a revised version of the manuscript that addresses the points raised during the review process.

**ACADEMIC EDITOR: I agree with the reviewer.**
**Strengths of your manuscript:**
**Weakness**
**1.**

We look forward to receiving your revised manuscript.

Kind regards,

Gauri Mankekar, MD,PhD,FACS

Academic Editor

PLOS ONE

3. In the online submission form, you indicated that [The data underlying the results presented in the study are available from corresponding author.].

Additional Editor Comments (if provided):

Reviewers' comments:

Reviewer's Responses to Questions

**Comments to the Author**

1. Is the manuscript technically sound, and do the data support the conclusions?

Reviewer #1: No

2. Has the statistical analysis been performed appropriately and rigorously?

Reviewer #1: N/A

3. Have the authors made all data underlying the findings in their manuscript fully available?

Reviewer #1: Yes

4. Is the manuscript presented in an intelligible fashion and written in standard English?

Reviewer #1: Yes

Reviewer #1: Thank you for the opportunity to review this manuscript. The research topic is important and highlights how the interpretation of AI answers differs between specialty doctors and non-specialized doctors. But there are several methodological concerns that significantly limit the strength of the findings.

First, an introduction to the abstract is not written.

For the methodology:

1- What was the rationale for specifically using the 10 questions presented.

2- The sample size calculation is not mentioned.

3- The used Likert scale was not tested for reliability and validity. Adding the binary question does not justify not testing the reliability and validity. So was the Likert scale validated in another study?

4- Adding the reasons behind "yes" answers for whether there was anything clinically wrong in each ChatGPT-4o response would have given better context to understand, especially the pulmonologist reasons.

Having all those statistically significant results without a clear sample size calculation, and without knowing if the study had enough/over power or not, along with not testing the reliability and validity of the Likert scale, means that even if p-values are <0.05, you can’t be sure the scale measured what it was supposed to.

**Do you want your identity to be public for this peer review?** For information about this choice, including consent withdrawal, please see our Privacy Policy

Reviewer #1: No

---

## [Author Response · Author response to Decision Letter 1]

5 Dec 2025

Dear Editor and Reviewers,

We sincerely thank you for the thoughtful and constructive feedback provided on our manuscript titled “Evaluation of the reliability and risks of ChatGPT-4o in answering pediatric cough questions: a comparative analysis between pediatricians and pediatric pulmonologists.” We carefully considered each comment and revised the manuscript accordingly. Below we provide a point-by-point response, detailing how each concern was addressed.

We believe these revisions have substantially strengthened the clarity, methodological rigor, and interpretability of our study.

EDITOR COMMENTS

Editor Comment 1

“Study design lacks systematic methodology and validation. The design does not include open-ended questions which can help understand the limitations of ChatGPT responses.”

Response:

We agree with the editor’s concern. We added a justification for not including open-ended items and acknowledged this as a study limitation. Specifically:

• Methods → Study Design: We explained that open-ended questions were considered but excluded to minimize participant burden and maintain response rates in a geographically dispersed physician sample.

• Discussion → Limitations: We now explicitly state that the absence of qualitative explanations limits interpretation and that future studies should incorporate open-ended follow-up questions.

Editor Comment 2

“Authors should discuss why ChatGPT was selected for this study when several AI models exist.”

Response:

A new paragraph was added in the Methods section explaining why ChatGPT-4o was selected, emphasizing its widespread use, public accessibility, multilingual capacity, and performance in recent medical evaluation studies.

Editor Comment 3

“Statistical analysis: The study is underpowered to detect small differences. Suggest adding effect sizes for all comparisons; report power analysis; include 95% CIs; include boxplots for Likert ratings.”

Response:

All recommended improvements were implemented:

• Effect sizes (r) and 95% confidence intervals were added for all Mann–Whitney U analyses (Tables 1 and 2).

• A post hoc power analysis using G*Power 3.1 was conducted and reported in the Statistical Analysis section.

• Boxplots illustrating the distribution of Likert ratings for both specialties were added as Figure 1.

Editor Comment 4

“Discussion: Discuss pulmonologists’ responses and whether associated with case complexity; discuss study limitations such as subjective Likert ratings.”

Response:

We expanded the Discussion to include:

• A detailed explanation of why pediatric pulmonologists provided more critical ratings (complex cases, rare presentations, greater sensitivity to nuance).

• An explicit acknowledgment of Likert-scale subjectivity and the absence of formal psychometric validation.

• An addition on content/face validity established through expert review.

REVIEWER COMMENTS

Reviewer Comment 1

“An introduction to the abstract is not written.”

Response:

We revised the abstract to include a clear introductory sentence that establishes the rationale behind the study.

Reviewer Comment 2

“What was the rationale for specifically using the 10 questions presented?”

Response:

We added a statement explaining that the selected questions consistently appeared across repeated unbiased Google searches and represent the most common parent-driven concerns in pediatric respiratory practice.

Reviewer Comment 3

“The sample size calculation is not mentioned.”

Response:

A post hoc power analysis was conducted and added as recommended, showing adequate power (>80%) for the observed effect sizes.

Reviewer Comment 4

“The Likert scale was not tested for reliability and validity.”

Response:

We now provide a detailed methodological explanation:

• The four Likert items intentionally measured distinct constructs, so internal consistency metrics (e.g., Cronbach’s alpha) were not appropriate.

• However, content and face validity were ensured through expert review by two pediatric pulmonologists.

• The absence of formal psychometric validation is explicitly discussed as a study limitation.

Reviewer Comment 5

“Adding the reasons behind ‘yes’ answers would have given better context.”

Response:

We fully agree. We explicitly acknowledged this as a limitation and clarified that lack of qualitative explanations restricts interpretability. We also recommended including open-ended follow-ups in future research.

We thank the editor and reviewers for their valuable feedback. We believe the revisions have substantially improved the quality, methodological transparency, and clarity of the manuscript. We hope the revised version meets your expectations and look forward to your favorable consideration.

Sincerely,

The Authors

---

## [Decision Letter · Decision Letter 1]

15 Dec 2025

Evaluation of the reliability and risks of ChatGPT-4o in answering pediatric cough questions: a comparative analysis between pediatricians and pediatric pulmonologists

PONE-D-25-38006R1

Dear Hanife Tuğçe Çağlar,

We’re pleased to inform you that your manuscript has been judged scientifically suitable for publication and will be formally accepted for publication once it meets all outstanding technical requirements.

Kind regards,

Gauri Mankekar, MD,PhD,FACS

Academic Editor

PLOS One

Additional Editor Comments (optional):

Reviewers' comments:

Reviewer's Responses to Questions

**Comments to the Author**

Reviewer #1: All comments have been addressed

2. Is the manuscript technically sound, and do the data support the conclusions?

Reviewer #1: Yes

3. Has the statistical analysis been performed appropriately and rigorously?

Reviewer #1: Yes

4. Have the authors made all data underlying the findings in their manuscript fully available?

Reviewer #1: Yes

5. Is the manuscript presented in an intelligible fashion and written in standard English?

Reviewer #1: Yes

Reviewer #1: (No Response)

**Do you want your identity to be public for this peer review?** For information about this choice, including consent withdrawal, please see our Privacy Policy

Reviewer #1: No

---

## [Editor Report · Acceptance letter]

PONE-D-25-38006R1

PLOS One

Dear Dr. Çağlar,

I'm pleased to inform you that your manuscript has been deemed suitable for publication in PLOS One. Congratulations! Your manuscript is now being handed over to our production team.

Kind regards,

on behalf of

Dr. Gauri Mankekar

Academic Editor

PLOS One